# Deep Fully Convolutional Network for MR Fingerprinting

**Dongdong Chen**[1]                                                    D.CHEN@ED.AC.UK
**Mohammad Golbabaee**[2]                                 M.GOLBABAEE@BATH.AC.UK
**Pedro A. Gómez**[3,4]                                          PEDRO.GOMEZ@TUM.DE
**Marion I. Menzel**[3,4]                                              MENZEL@GE.COM
**Mike E. Davies**[1]                                            MIKE.DAVIES@ED.AC.UK

[1] *School of Engineering, The University of Edinburgh*

[2] *Computer Science department, University of Bath*

[3] *School of Bioengineering, Technische Universität München*

[4] *GE Healthcare*

## Abstract

This work proposes an end-to-end deep fully convolutional neural network for MRF reconstruction (MRF-FCNN), which firstly employs linear dimensionality reduction and then uses a neural network to project the data into the tissue parameters. The MRF dictionary is only used for training the network and not during image reconstruction. We show that MRF-FCNN is capable of achieving accuracy comparable to the ground-truth maps thanks to capturing spatio-temporal data structures without a need for the non-scalable dictionary matching step used in the baseline reconstructions.

**Keywords:** Magnetic Resonance Fingerprinting, Deep Learning

## 1. Introduction

Magnetic Resonance Fingerprinting (MRF) (Ma et al., 2013) has emerged as a promising quantitative Magnetic Resonance Imaging approach, which can significantly reduce the acquisition time needed for quantitative measurements. However, the conventional MRF methods suffer from the heavy storage and computation requirements of a dictionary-matching (DM) step due to the growing size and complexity of the fingerprint dictionaries in multi-parametric quantitative MRI applications (Davies et al., 2014). Recently, many deep learning based MRF reconstructions (Hoppe et al., 2017; Cohen et al., 2018) have been proposed to exploit deep neural networks to replace the dictionary and the lookup-table used in conventional MRF reconstruction approaches. Alternately, by imposing a linear dimension reduction procedure, our proposed MRF-Net (Golbabaee et al., 2019) is able to accurately approximate the DM step saving more than 60 times in memory and computations. This paper extends the learning capability of the MRF-Net by including a fully convolutional architecture that is capable of capturing both spatial and temporal structures.

## 2. Proposed Method

The proposed MRF-FCNN consists of two components: a linear projector $\mathcal{P}_0 : \mathbb{R}^{m \times d_0} \to \mathbb{R}^{m \times d_1}$ and a neural network projector $\mathcal{P}_1 : \mathbb{R}^{m \times d_1} \to \mathbb{R}^{m \times d_2}$, where $m$ is the number

of voxels, $d_0$ is the number of acquired time points (i.e., dimension of each fingerprint), $d_1 = 10$ is the reduced dimensionality, and $d_2 = 3$ corresponding to the desired tissue's intrinsic parameters e.g., $\Theta = [T1, T2, PD]$ where $T1, T2$ are relaxation times and $PD$ denotes proton density. The former aims to learn a linear projection onto the subspace of clean fingerprints. The latter devotes to nonlinearly project the dimension-reduced data onto the manifold of $\Theta$, due to the powerful ability of neural networks for manifold learning (Chen et al., 2017, 2018b) and pattern recognition (Chen et al., 2018a; Li et al., 2019). Therefore, MRF-FCNN finally approximates the following transformation $\mathcal{F}$:

$$\mathcal{F} : \mathcal{P}_0 \circ \mathcal{P}_1. \tag{1}$$

In this work, we apply principal component analysis (PCA) as the $\mathcal{P}_0$ and a concisely designed fully convolutional neural network as the MRF-FCNN (Figure 1). It starts with an unsupervised learning layer (gray) which learns a linear projection onto the subspace of clean fingerprints through PCA, then keep $\mathcal{P}_0$ fixed during the training of the other layers. The main component (dotted box) of the MRF-FCNN is designed with stacks of separable convolutional layers (yellow) with kernel size $3 \times 3$, and decreasing feature maps $(256, 128, 64, 32)$ for fine texture features learning, and finally ends with two convolutional layers (green) with the same kernel size $1 \times 1$ and 3 feature maps for each layer. The ReLu are used as the activation function, as it provides a piece-wise affine approximation to the Bloch response manifold projection (Golbabaee et al., 2019) (i.e. a transformation from tissue's intrinsic parameters $[T1, T2]$ to it's corresponding temporal signature). The followed dropouts are included to prevent over-fitting. Our empirical studies show that including the $1 \times 1$ convolutional layers at the end of the model is crucial to the reconstruction, which reduces the redundancy in former feature maps and increases the correlation between channels and finally helps to prevent local blurring in our MRF reconstruction.

Our approach performs a sharp and accurate parameter estimation. The proposed MRF-FCNN uses spiral sub-sampled MRF data but it reconstructs the data with similar accuracy to the Cartesian sampled images acquired using a specific protocol, e.g. MAGIC (Marcel and AB, 2015). In addition, a benefit from the dimensionality-reduction operator, the MRF-FCNN requires far less units and training resource which distinguishes from other mainstream deep learning approached applied to MRF (Hoppe et al., 2017; Cohen et al., 2018), and our experimental results show that the MRF-FCNN does not suffer from common blurring artifacts in spiral sampling protocols.

## 3. Results

We test the MRF-FCNN on a simulated human brain MRF data. To be specific, the ground truth used for this simulation was acquired using longer protocol MAGIC (Marcel and AB, 2015). In this work, we set the image scale $m = 256 \times 256$, and we collect ground truth (GT) parametric maps from 8 volunteers (20 brain slices each) using MAGIC quantitative MRI protocol with Cartesian sampling. These parametric maps are then used to simulate MRF acquisition using the Fast imaging with Steady State Precession (FISP) (Jiang et al., 2015) protocol and spiral sampling. Accordingly, the input to MRF-FCNN are the Fourier back-projected images corresponding to the MRF $k$-space measurements. These are highly aliased due to operating in a sever under-sampling regime (i.e. only 732 $k$-space samples are

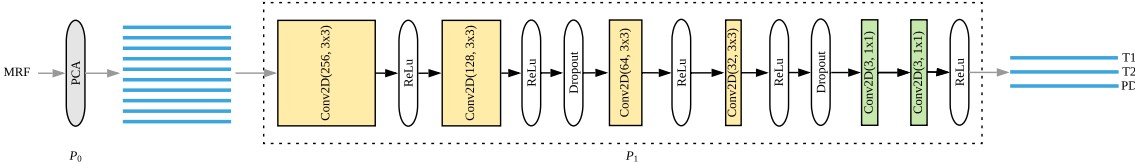

Figure 1: An illustration of the proposed MRF-FCNN. Inputs are the voxel sequences and output are the per-voxel T1, T2 and PD parameters.

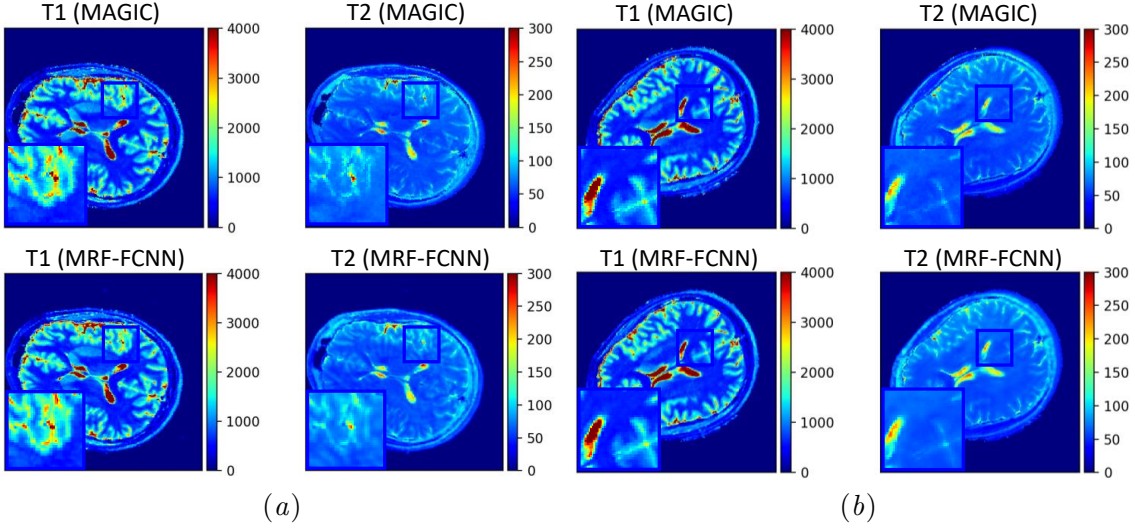

Figure 2: Two examples of the reconstructed T1 and T2 maps using the proposed MRF-FCNN. For each example (a) and (b), the first row of images are the ground truth parametric maps acquired using MAGIC protocol with Cartesian sampling, the bottom images are our MRF reconstruction results using FISP protocol and spiral sampling.

collected at each time-point). The output are the GT parametric maps. To avoid overfitting, the standard data augmentation is conducted by adding the translation, rotation, scale, and noise. In our experiment, the first 7 volunteers' data are used for training while the data of last volunteer are for reconstruction testing. We train the MRF-FCNN on two Nvidia 1080Ti GPUs. The reconstructed maps (Figure 2) indicate the proposed MRF-FCNN could generate high-quality reconstruction similar to MAGIC without suffering from blurring artifacts. More importantly, the standard dictionary matching (search) approaches would typically take a couple of minutes for reconstruction, but MRF-FCNN only takes around 0.29 seconds for a single slice MRF reconstruction. This means we can use the MRF-FCNN framework and get similar quality quantitative images as MAGIC but in a much shorter acquisition time. Detailed comparison with other methods in terms of reconstruction quality, computational performance and applicability to real-world data will be addressed in future work.

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
