# OpenReview forum: "Deep Fully Convolutional Network for MR Fingerprinting"
_MIDL.io/2019/Conference/Abstract — MIDL Abstract 2019_

### Official Review · AnonReviewer1 · 2019-04-29
**Limited validation but very interesting methodology and application area**

**Rating:** 3
**Confidence:** 2

**Review:**

This paper present a reconstruction framework for magnetic resonance fingerprinting using a convolutional neural network. Although the validation is limited to a single dataset with seven datasets for testing, the premise is interesting and the methodology clearly described.

---

### Official Review · AnonReviewer2 · 2019-05-01
**no experiment setup details are given**

**Rating:** 2
**Confidence:** 2

**Review:**

There is no experiment setup details are given so it is hard to judge the effectiveness of the proposed approach, on simulated or real patient data? If real patient data were used, how many patients and what is the patient study recruiting protocol?

---

### Decision · Program_Chairs · 2019-05-06
**Acceptance Decision**

Accept